REGISTERED REPORT PROTOCOL

# Addressing the immigrant screening gap: A protocol for a systematic review on interventions to enhance colorectal cancer screening among immigrants in the United States

**Taylor M. McCready** [ID][1]*, **Ethan M. Cohen**[2], **Gregory Laynor** [ID][3], **Perla Chebli**[1], **Peter S. Liang**[1,4,5], **Audrey Renson**[1]

1  Department of Population Health, New York University Grossman School of Medicine, New York, New York, United States of America, 2  Section of Internal Medicine, West Virginia University, West Virginia, West Virginia, United States of America, 3  Health Sciences Library, New York University Grossman School of Medicine, New York, New York, United States of America, 4  Department of Medicine, New York University Grossman School of Medicine, New York, New York, United States of America, 5  Department of Medicine, Veterans Affairs New York Harbor Healthcare System, New York, New York, United States of America

* taylor.mccready@nyulangone.org (TM)

## Abstract

### Introduction

Colorectal cancer (CRC) screening rates are lower among immigrant populations in the United States (US) than the general population. Immigrant communities face structural barriers that disincentivize their engagement from CRC screening. A growing body of literature has evaluated the effects of interventions aimed at increasing CRC screening engagement among various immigrant groups, but there has not yet been a systematic synthesis of this literature.

### Objective

This review will systematically evaluate quantitative studies assessing the effects of interventions designed to increase CRC screening rates among immigrant populations residing in the US.

### Methods

We will conduct a comprehensive search of English language peer-reviewed and grey literature using specific keywords and database-specific structured vocabulary on interventions to improve CRC screening rates among immigrants published in 7 databases (PubMed, Cochrane Library (Wiley), CINAHL (EBSCO), ClinicalTrials. gov, Embase (Ovid), Scopus (Elsevier), and Web of Science) from January 1, 2000 to December 31, 2024. All studies will be imported into Covidence. Two reviewers will independently screen titles, abstracts, and full-texts for inclusion and exclusion

This is a Registered Report and may have an associated publication; please check the article page on the journal site for any related articles.

**Data availability statement:** No datasets were generated or analyzed during the current study, as this is a protocol. All relevant data from this study will be made available upon study completion.

**Funding:** The author(s) received no specific funding for this work.

**Competing interests:** Peter S. Liang reports research support from Freenome and consulting for Guardant Health and Natera. No other authors have anything to disclose. This does not alter our adherence to PLOS ONE policies on sharing data and materials.

criteria. Pilot screenings and consensus discussions will ensure accuracy and agreement in study selection and data extraction. Iterative data extraction of eligible studies will include critical appraisal using the Risk of Bias 2 (ROB2) for randomized controlled trials, while other study designs will be appraised with the risk of bias in nonrandomized studies of interventions (ROBINS-I) tool. Data synthesis will disaggregate pooled effect estimates by ethnicity, to the extent possible. The study protocol was pre-registered in International Prospective Register of Systematic reviews (PROSPERO): CRD42023488183.

## Expected outputs

This systematic review aims to generate an exhaustive summary of the evidence base, including a description of the intervention methods and settings, target populations, recruitment and retention strategies, partnerships and collaborations, and reported outcomes. The results will provide actionable recommendations for public health practitioners, healthcare providers, and policymakers developing tailored interventions and policies aimed at improving CRC screening uptake among diverse immigrant populations in the US.

---

## Introduction

Colorectal cancer (CRC) is the third most common cancer diagnosis in the United States (US) among men and women, accounting for 7.9% of all new cancer cases [1]. CRC screening is a crucial preventive service, estimated to account for roughly half of the observed reduction in CRC incidence and mortality in the US in the era of increased colonoscopy screenings from 1975–2010 [2]. Yet, on average, only 59% of Americans older than 45 are up to date with screening [3,4].

This shortfall in screening rates is particularly pronounced among immigrant populations, who make up 14.9% of the US population, totaling over 48 million people in 2024 [5]. Immigrants who have lived in the US for less than 10 years and are over 45 years old exhibit substantially lower screening rates—less than half (30%) compared to their US-born counterparts (60%) [6]. This disparity underscores a critical gap in the effectiveness of existing public health campaigns, which often overlook the specific needs (e.g., health literacy, insurance access) of communities disproportionately affected by inadequate screening access and uptake [7].

In particular, screening-related disparities are likely driven by structural racism [8]. The intricate connection between immigration status and racial identity is deeply rooted in the US, evidenced by current and past immigration policies designed to intentionally bar certain groups from gaining citizenship and its related benefits [7]. These exclusions are achieved not only by limiting citizenship to those identifying as 'white,' but also by creating specific racial classifications for separate groups [7–10]. The downstream effects of structural racism impede cancer screening by creating substantial systemic barriers. More specifically, there is stigma associated with lack of insurance (e.g., non-citizens must wait 5 years after receiving "qualified"

immigration status before they are eligible for Medicaid and CHIP coverage), low-cost healthcare settings are often subject to high turnover or poor care, resulting in fragmented and discontinuous care, and low-wage jobs typically lack benefits that do not offer the flexibility required to attend doctor's appointments, especially for preventative purposes [11,12]. Despite the fact that immigrants in the US report common experiences of racialized discrimination that appear to impact long-term health outcomes [7,13,14], cultural explanations often dominate discussions about immigrant health disparities [7,10]. Culturally-tailored interventions are extremely valuable, as each immigrant group brings distinct histories and experiences. However, since structural racism affects all racialized immigrant groups, Misra et al. (2022) argue that interventions can also be structured to utilize these similarities instead of solely concentrating on the unique cultural requirements of particular immigrants communities [7]. This approach can inform more inclusive and sustainable interventions without undermining the value of culturally tailored strategies [7,15].

A growing body of literature has evaluated the effects of interventions aimed at increasing CRC screening engagement among various immigrant groups. Typically, these are either educational sessions aimed at increasing awareness and reasoning behind CRC screening or culturally tailored interventions with patient navigators [16,17]. However, there has not been a systematic synthesis of these studies. Though systematic reviews of interventions to improve screening rates among predominantly non-immigrant populations exist, reviews focused on immigrant populations have only considered intermediate outcomes (e.g., awareness, barriers) [18–22], were limited to immigrants of a specific nationality (in this case, Somali) [23], or were non-systematic and devoted limited attention to CRC in the context of cancer screening more broadly [24]. Similarly, a mini-review highlighted the progress in promoting cancer screening participation among US immigrants, but was not a systematic review, was restricted to a single database, and concentrated primarily on breast and cervical cancer screening interventions rather than colorectal cancer [24]. In this paper, we describe the protocol for a systematic review of interventions to increase CRC screening rates among immigrants, defined broadly [25,26], using the Cochrane method.

## Materials and methods

### Stage 1: Identifying the research question

**Objective. Main research question.** This review will identify and evaluate the effectiveness of interventions aimed at increasing the uptake of CRC screening among immigrants in the US.

Specifically, our objectives are:

Aim 1: Conduct a comprehensive analysis of CRC screening interventions for immigrants

   1a.  Identify and narratively describe interventions targeting CRC screening uptake among immigrants in the US

   1b.  Identify gaps in the current literature

Aim 2: Evaluate intervention effectiveness and research quality

   2a.  Summarize effect estimates of the most common interventions, both overall and disaggregated by country or region of origin

   2b.  Assess the quality of, and the risk of bias in the identified quantitative evidence

**Protocol and registration.** The protocol is reported in accordance with the Preferred Reporting Items for Systematic Review and Meta-Analysis Protocols (PRISMA-P) 2020 guidelines (see PRISMA-P Checklist in S2 File) [27]. The protocol was registered with PROSPERO: CRD42023488183. Amendments to the systematic review protocol, if applicable, will be documented on our PROSPERO registration page.

### Stage 2: Study eligibility.

**Inclusion criteria.** An article will be included if it satisfies all of the following conditions (AND criteria):

1. Was published in the English language and in the US

2. Is a peer-reviewed quantitative research study of any design, demonstrating a clear measurement of an intervention to improve the completion or uptake of any recommended CRC screening test (e.g., fecal occult blood test, sigmoidoscopy, colonoscopy) as confirmed either by self-report or medical records.

3. Is a study with immigrants from any background

4. Is a study published or a clinical trial completed between 2000 through June 30, 2024

The term 'immigrant' lacks a standardized definition in academic research. Some studies define immigration status by nativity or number of years spent in the US, while others define it generationally (e.g., first generation, second generation, 1.5 generation), culturally (e.g., Korean-American, Nigerian-American) or by documentation status [28]. For the purposes of this study, we will use the definition of immigrant provided by the International Organization on Immigration which defines an immigrant, *"from the perspective of the country of arrival, a person who moves into a country other than that of his or her nationality or usual residence, so that the country of destination effectively becomes his or her new country of usual residence"* [29–31].

## Exclusion criteria

An article will be excluded if it

1. Reports only intermediate outcomes such as knowledge, attitudes, or intentions – with no data on CRC screening behavior (e.g., receipt of a CRC screening test)

2. Is a descriptive study that only investigates patterns of colorectal cancer screening uptake among immigrant populations

3. Contains only research conducted outside of the US, or

4. Is a systematic review or other type of review article; however, the references within such reviews will be examined and considered for inclusion if they meet the established criteria.

Studies conducted solely outside of the US will be considered ineligible given the unique context of the US immigration and healthcare system, although we will include studies conducted in multiple countries inclusive of the US if results are reported by country. We will also exclude articles that are conference abstracts or study protocols if they do not provide enough data to extract, are missing full text records, or don't have results posted on ClinicalTrials.gov.

## Stage 3: Search strategy

The review team will work with a research librarian (GL) to develop a systematic review search strategy. Search terms to define immigrant, CRC and CRC screening interventions will be exhaustive of terms we identify in past studies [18,32]; preliminary scoping searches will be conducted in MEDLINE via PubMed to iteratively identify search terms. The search strategy will combine Medical Subject Headings (MeSH) terms and keywords as well as input from subject matter experts (PL, GL) [31]. Changes to the search will be documented. Polyglot search translator will be used to translate the MEDLINE PubMed search syntax to the syntax for additional databases [33]. The translated syntax will be manually revised by two authors (TM, GL) to ensure accuracy and consistency with database-specific structured vocabulary.

We will apply the search strategy to PubMed, Cochrane Library (Wiley), CINAHL (EBSCO), Embase (Ovid), Scopus (Elsevier), and Web of Science. ClinicalTrials.gov will also be searched for completed trials not yet published. TM will enter records into the EndNote reference management software and perform automated deduplication [34]. The proposed PubMed syntax is presented in Table 1 and the corresponding translated search strategies are included in Table A in S1 File.

## Stage 4: Selection of sources of evidence and data collection

**Data management.** We will use Covidence, an online collaborative tool for managing and streamlining reviews, for screening titles/abstracts, full text review, data extraction, and risk of bias assessments [35]. Once references are entered into Covidence, two independent reviewers (TM, EC) will conduct blinded screening of titles and abstracts to independently ensure that the documents fulfil the inclusion and exclusion criteria. Studies that appear to be relevant will be earmarked for a detailed full-text review. To ensure a high level of consensus and refine the screening process, we will undertake a pilot screening of 30 titles and abstracts to establish inter-rater reliability. Inclusion and exclusion criteria will be revised as necessary for clarification during the pilot. We will assess inter-rater reliability for all screened titles and abstracts using Cohen's kappa coefficient and will consider 0.60 or higher as an acceptable level of inter-rater reliability [36].

After title and abstract screening, both reviewers (TM, EC) will review all full texts to assess for inclusion in the review as per eligibility criteria. A list of disagreements will be resolved through discussion with a third reviewer (AR), who will adjudicate the final inclusion decision. Similarly, for full-text reviews, a pilot screening will be executed on 10% of the selected articles. This two-tier pilot approach will aid in calibrating the review process and minimizing discrepancies. A list of titles excluded at this stage, as well as the reason for their exclusion, will be included in an appendix in the final paper.

**Table 1. Proposed PubMed search syntax.**

| Search Terms | | |
| --- | --- | --- |
| Immigrant | | (((((((((("refugee*"[Title/Abstract])) OR ("migrant*"[Title/Abstract])) OR ("emigrants and immigrants"[MeSH Terms])) OR ("immigrant*"[Title/Abstract])) OR ("Transients and Migrants"[MeSH Terms])) OR ("emigrant*"[Title/Abstract])) OR ("immigration"[Title/Abstract])) OR ("foreign born"[Title/Abstract])) |
| Colorectal Cancer | AND | ((((((((((((((((((("colorectal cancer*"[Title/Abstract] OR("colorectal neoplasms"[MeSH Terms])) OR ("colorectal neoplasm*"[Title/Abstract])) OR ("neoplasm colorectal"[Title/Abstract])) OR ("colorectal tumors"[Title/Abstract])) OR ("tumor colorectal"[Title/Abstract])) OR ("cancer colorectal"[Title/Abstract])) OR ("colorectal carcinoma*"[Title/Abstract])) OR ("carcinoma* colorectal"[Title/Abstract])) OR ("Colonic Neoplasms"[MeSH Terms])) OR ("colorectal adenocarcinoma*"[Title/Abstract])) OR ("colon cancer*"[Title/Abstract])) OR ("rectum cancer*"[Title/Abstract])) OR ("rectal cancer*"[Title/Abstract])) OR ("adenocarcinoma* colon"[Title/Abstract])) OR ("mucinous adenocarcinoma of the colon"[Title/Abstract])) OR ("signet ring adenocarcinoma of the colon"[Title/Abstract])) OR ("Adenocarcinoma rectum"[Title/Abstract])) OR ("squamous cell carcinoma rectum"[Title/Abstract])) OR ("squamous cell carcinoma rectal"[Title/Abstract])) OR ("Adenocarcinoma rectal"[Title/Abstract])) OR ("cancer of rectum"[Title/Abstract])) OR ("cancer of colon"[Title/Abstract]))) |
| Screening | AND | (((((((((((((((((((((((((((((((((((("colonoscopy*"[Title/Abstract] OR ("colonoscopy"[MeSH Terms]))) OR ("colonoscopies"[Title/Abstract])) OR ("Colonoscopic Surgical Procedures"[Title/Abstract])) OR ("Colonoscopic Surgical Procedures"[Title/Abstract])) OR ("Procedure Colonoscopic Surgical"[Title/Abstract])) OR ("Procedures Colonoscopic Surgical"[Title/Abstract])) OR ("Surgical Procedure Colonoscopic"[Title/Abstract])) OR ("Surgery, Colonoscopic"[Title/Abstract])) OR ("Surgical Procedures, Colonoscopic"[Title/Abstract])) OR ("Colonoscopic Surgery"[Title/Abstract])) OR ("Colonoscopic Surgeries"[Title/Abstract])) OR ("Surgeries, Colonoscopic"[Title/Abstract])) OR ("sigmoidoscopy*"[Title/Abstract])) OR ("sigmoidoscopy"[MeSH Terms])) OR ("Sigmoidoscopies"[Title/Abstract])) OR ("Proctosigmoidoscopy"[Title/Abstract])) OR ("proctosigmoidoscopies"[Title/Abstract])) OR ("Sigmoidoscopic Surgical Procedures"[Title/Abstract])) OR ("Procedure Sigmoidoscopic Surgical"[Title/Abstract])) OR ("Procedures Sigmoidoscopic Surgical"[Title/Abstract])) OR ("Sigmoidoscopic Surgical Procedure"[Title/Abstract])) OR ("Surgical Procedure Sigmoidoscopic"[Title/Abstract])) OR ("Surgery Sigmoidoscopic"[Title/Abstract])) OR ("Surgical Procedures Sigmoidoscopic"[Title/Abstract])) OR ("Sigmoidoscopic Surgery"[Title/Abstract])) OR ("Sigmoidoscopic Surgeries"[Title/Abstract])) OR ("Surgeries Sigmoidoscopic"[Title/Abstract])) OR ("FOBT"[Title/Abstract])) OR ("Fecal occult blood test"[Title/Abstract])) OR ("Occult Blood"[MeSH Terms])) OR ("Stool test"[Title/Abstract])) OR ("Screen*"[Title/Abstract])) OR ("early detect"[Title/Abstract])) OR ("early diagnosis"[Title/Abstract]))) OR ("early detection of cancer"[Title/Abstract])) OR ("screening rates"[Title/Abstract])) OR ("screening rate"[Title/Abstract])) OR ("FIT"[Title/Abstract])) OR ("fecal immunochemical test"[Title/Abstract])) OR ("mt-sDNA"[Title/Abstract])) OR ("multi-target stool DNA test"[Title/Abstract])) OR ("CT colonography"[Title/Abstract])) OR ("Computer tomographic colonography"[MeSH Terms])) OR ("CT colonography"[MeSH Terms])) OR ("Colonography, CT"[MeSH Terms])) OR ("Virtual colonoscopy"[MeSH Terms])) OR ("Colonoscopy, Virtual"[MeSH Terms]) |

The asterisk wildcard symbol (*) is used to find variants of word endings.

MeSH, Medical Subject Headings.

If multiple included records pertain to a single study, we will group these records for extraction. The PRISMA flow chart will be used to document the total number and reasons for exclusion at each stage (S2 Table).

## Data collection

Our data extraction process is designed to accommodate the diverse study designs and population characteristics anticipated in our review. The extraction tool, detailed in Table B in S1 File and adapted from the Cochrane data collection form, is optimized for use with Covidence (Table B in S1 File) [26]. In accordance to Cochrane recommendations, data will be extracted independently by two authors. We will pilot test our data extraction approach using 10% of articles that meet full-text criteria and iteratively adjust the data extraction instrument as needed. One reviewer (TM) will review data entries for completeness and accuracy utilizing Covidence's consensus feature [35]. If consensus cannot be reached after discussion with both reviewers, the supervising author will be consulted (AR). If clarifications of published data are needed (e.g., sample size, reported screening rates), we will attempt to contact the corresponding author once by email at the address provided in the publication [26].

We will extract article and study information including the first author, publication year, study design, and study years. We will also extract information on target population characteristics, informed by National Institutes of Health-designated US health disparity populations and previous reviews [37]. This information will encompass baseline population demographics such as nativity, years in the US, language, citizenship, and insurance status, which are crucial for an in-depth discussion about generalizability and equity potential. The primary outcome data extracted will be the raw numbers of screened and unscreened subjects in each arm and/or effect estimate, but additional, extensive outcome, intervention, and study characteristics will be collected for each study including type of screening method offered (FOBT, colonoscopy, sigmoidoscopy), pre- and post-intervention screening rates, study setting (community center, church, outpatient medical appointment) and detailed characterization of each intervention including recruitment strategies, partnerships with community organizations, and implementers (i.e., patient navigators, physicians).

### Stage 5: Data synthesis.

**Risk of bias.** To assess the risk of bias in quantitative studies included within the review, the Risk of Bias 2 (ROB2) and Risk of Bias In Non-randomized Studies of Interventions (ROBINS-1) tools will be used [26,38–40]. The ROB2 tool was designed to assess the risk of bias in randomized trials along five domains: bias arising from the randomization process, bias due to deviations from intended interventions, bias due to missing outcome data, bias in measurement of the outcome, and bias in the selection of the reported result [40]. The Risk of Bias in Nonrandomized Studies of Interventions (ROBINS-I) tool will be used for all other observational study designs (e.g., pre-post study designs, cohort studies, cross-sectional) [38]. This tool was developed to assess 7 domains of bias (e.g., confounding, selection of participants, measurement of the outcome) in nonrandomized studies recognizing that they can provide important contributions but are also prone to bias due to their non-randomized design [38]. In both tools, signaling questions are designed to help assess bias risk for each domain-level judgements about risk of bias, and ultimately, make an overall risk of bias judgement for each particular outcome [38,40].

Similarly to the data extraction tools, the risk of bias assessments will be built in Covidence. Two reviewers (TM, EC) will apply the relevant tool during data extraction, resolving disagreements by discussion or, if necessary, with the supervising author (AR). We will report on themes that contribute to a higher risk of bias rating in the full review (i.e., unmasked outcome assessment, differential awareness of trial participation).

## Ethics

We are not collecting and analyzing primary data; rather, we are reviewing and extracting information from already published studies. Therefore, IRB approval is not required.

## Data analysis

Due to the expected heterogeneity in studies, we will first conduct a narrative synthesis of data from included studies based on intervention type; intervention types may include patient navigation with or without culturally tailored materials; reminder outreach via mail, phone, or text message; and financial incentives. We will generate a table of study characteristics including design, time to assessment, participants, intervention category, study setting, comparison groups, results, and risk of bias rating. We will further break down interventions by country or region of origin as available, as well as provide a synthesis on studies that state they include immigrants as part of a broader sample, but do not report results stratified by immigration status.

If appropriate, data will be synthesized in a meta-analysis, but to date, we do not intend to conduct a formal meta-analysis, given the anticipated heterogeneity in study designs and backgrounds of immigrant groups. If a meta-analysis is not possible, we will consider using an alluvial chart to display the qualitative findings [41]. For studies that do not have a comparison group, changes in screening use before and after the intervention will be calculated. If a study includes multiple outcome time points, the screening at the longest follow-up will be used to establish post-intervention screening use. For trials with multiple arms, we will assess the outcomes of all active interventions vs standard of care and vs other active comparators. Secondary outcomes unrelated to the objectives will not be synthesized [26].

## Discussion

This systematic review aims to identify and evaluate existing research on interventions targeting immigrant communities to enhance CRC screening rates. Evidence-informed clinical guidelines exist for CRC screening in the general population, and implementation of these has been shown to lead to improved survival outcomes and a decreased burden on the US healthcare system [2]. However, racialized and ethnicized inequalities persist in screening uptake [6,12]. By systematically synthesizing the available evidence, this review will shed light on effective strategies, identify gaps in current research, and inform the development of targeted interventions for diverse immigrant populations. Highlighting specific approaches tailored to address the unique barriers faced by immigrant communities will be instrumental in reducing disparities in CRC screening and addressing structural inequities within existing healthcare guidelines.

## Supporting information

**S1 File. Search string and data extraction template.**
(DOCX)

**S2 File.** PRISMA-P (Preferred Reporting Items for Systematic review and Meta-Analysis Protocols) 2020 checklist: Recommended items to address in a systematic review protocol.
(DOCX)

## Author contributions

**Conceptualization:** Taylor M McCready, Ethan M. Cohen, Gregory Laynor, Audrey Renson.

**Methodology:** Taylor M McCready, Ethan M. Cohen, Gregory Laynor, Audrey Renson.

**Supervision:** Gregory Laynor, Audrey Renson.

**Writing – original draft:** Taylor M McCready, Ethan M. Cohen, Gregory Laynor, Perla Chebli, Peter S. Liang, Audrey Renson.

**Writing – review & editing:** Taylor M McCready, Ethan M. Cohen, Gregory Laynor, Perla Chebli, Peter S. Liang, Audrey Renson.

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
