## [Decision Letter · Decision Letter 0]

14 Jan 2025

PONE-D-24-23935Addressing the immigrant screening gap: A protocol for a systematic review on interventions to enhance colorectal cancer screening among immigrants in the United StatesPLOS ONE

Dear Dr. McCready,

Thank you for submitting your manuscript to PLOS ONE. After careful consideration, we feel that it has merit but does not fully meet PLOS ONE’s publication criteria as it currently stands. Therefore, we invite you to submit a revised version of the manuscript that addresses the points raised during the review process.

We look forward to receiving your revised manuscript.

Kind regards,

Udoka Okpalauwaekwe, MD, MPH, PhD

Academic Editor

PLOS ONE

Journal Requirements:

2. In your cover letter, please confirm that the research you have described in your manuscript, including participant recruitment, data collection, modification, or processing, has not started and will not start until after your paper has been accepted to the journal (assuming data need to be collected or participants recruited specifically for your study). In order to proceed with your submission, you must provide confirmation.

“I have read the journal's policy and the authors of this manuscript have the following competing interests: Peter S. Liang reports research support from

Freenome and consulting for Guardant Health and Natera. No other authors have anything to disclose.”

Reviewers' comments:

Reviewer's Responses to Questions

**Comments to the Author**

1. Does the manuscript provide a valid rationale for the proposed study, with clearly identified and justified research questions?

Reviewer #1: Yes

Reviewer #2: Yes

2. Is the protocol technically sound and planned in a manner that will lead to a meaningful outcome and allow testing the stated hypotheses?

Reviewer #1: Yes

Reviewer #2: Partly

3. Is the methodology feasible and described in sufficient detail to allow the work to be replicable?

Reviewer #1: Yes

Reviewer #2: No

4. Have the authors described where all data underlying the findings will be made available when the study is complete?

Reviewer #1: Yes

Reviewer #2: No

5. Is the manuscript presented in an intelligible fashion and written in standard English?

Reviewer #1: Yes

Reviewer #2: Yes

6. Review Comments to the Author

You may also provide optional suggestions and comments to authors that they might find helpful in planning their study.

Reviewer #1: This is a very solid systematic review protocol. I have one suggestion regarding the data analysis and synthesis plan. Given the expected heterogeneity and the possibility of not conducting a meta-analysis, I recommend that the authors consider incorporating a visual illustration of their qualitative findings using alluvial charts. This method is a powerful way to report overall trends across all studies included in the systematic review. For reference, you may want to look at a recent example of this approach here: PMC9683573 (https://pubmed.ncbi.nlm.nih.gov/36417344/).

Reviewer #2: It was a pleasure reviewing this work which is a systematic review protocol to explore patterns and gaps in CRC screening. While the authors made a strong attempt in being rigourous in their delivery there are areas that need addressing to improve the rigour and quality of this work.

1)Firstly, the format for submission to PLOS ONE should be reviewed and adhered to. the referencing styles are mismatched (with superscripts and numbered styles used). The numbers are placed after the punctuation marks (which should be for superscripts). Eitherway stick to author guidelines and be consistent.

In the Eligibility criteria, provide further description for what will be considered screening behaviour to warrant exclusion. Also provide rationale for exclusion of review studies from inclusion. Would protocols be excluded too as you cite on line 214 "conducting searches forward from relevant, peer-reviewed research protocols to find articles discussing study results." Why is that?

What will be the grading system for consideration of moderate to good inter rater agreement for your Kappa scoring?

From Line 211 to 215, I don't understand the rational for the 3 steps itemized. I find it redundant if you already did that in your screening. Except it is any different from what is already described I advise further explaining your rationale or removing it. For example, why will you conduct searches again in step (c) to find articles discussing results? and what is the point of reviewing bibliographies again when you already mentioned it in your eligibility criteria on line 164.

What would you use to appraise cross-sectional studies. ROBINS-I?

Line 263: "If consensus cannot be reached after discussion with both reviewers, the supervising author will be consulted (AR)" is redundant. It has been described before in Line 204.

Line 264:"We will report on themes that contribute to a higher risk of bias rating in the full review (i.e., unmasked outcome assessment, differential awareness of trial participation)." what does this mean? what is the aim vis a vis your core objectives.

Line 228: "Missing data in publications that are requested on the extraction form will be elicited by contacting the listed contact author once by email at the email address listed on the publication." This is a bit vague and I am unsure if contacting corresponding authors would significantly benefit the design of your study, which relies on secondary data. If the purpose is to clarify already published data, this should generally be acceptable. However, if the intent is to collect additional data, such as missing information, this would likely require an ethics application for secondary use of data. I recommend confirming the ethical requirements with your institution to ensure compliance.

7. PLOS authors have the option to publish the peer review history of their article (what does this mean? ). If published, this will include your full peer review and any attached files.

**Do you want your identity to be public for this peer review?** For information about this choice, including consent withdrawal, please see our Privacy Policy .

Reviewer #1: **Yes: ** Dr. Jean C. Bikomeye

Reviewer #2: **Yes: ** Udoka Okpalauwaekwe MD, MPH, PhD

---

## [Author Response · Author response to Decision Letter 0]

14 Feb 2025

Title: Addressing the immigrant screening gap: A protocol for a systematic review on

interventions to enhance colorectal cancer screening among immigrants in the United States

Manuscript ID: PONE-D-24-23935

Date: 1/29/2025

Dear Editor and Reviewers,

We would like to thank you for your thorough reading of our manuscript and for the detailed, constructive feedback. We greatly appreciate the time and effort you have invested in reviewing our work. The suggestions and comments have helped us improve the clarity, rigor, and overall quality of our manuscript.

In what follows, we provide a detailed, point-by-point response to each comment. For clarity, we have listed each reviewer’s comment in bold followed by our response. All revisions made in the manuscript are indicated using tracked changes.

We hope that our responses and the accompanying revisions satisfactorily address your comments and concerns. Please let us know if there are any remaining questions or requests.

Sincerely,

Taylor M. McCready, MPH

Title: Addressing the immigrant screening gap: A protocol for a systematic review on

interventions to enhance colorectal cancer screening among immigrants in the United States

Manuscript ID: PONE-D-24-23935

Date: 1/29/2025

Point-by-Point Response

Reviewer #1

1.

Comment 1:

“This is a very solid systematic review protocol. I have one suggestion regarding the data analysis and synthesis plan. Given the expected heterogeneity and the possibility of not conducting a meta-analysis, I recommend that the authors consider incorporating a visual illustration of their qualitative findings using alluvial charts. This method is a powerful way to report overall trends across all studies included in the systematic review. For reference, you may want to look at a recent example of this approach here: PMC9683573 (https://pubmed.ncbi.nlm.nih.gov/36417344/).”

Response: If a meta-analysis is not possible, we will use an alluvial chart to display the qualitative findings. Thank you for this suggestion!

Reviewer #2

1.

Comment 1: Firstly, the format for submission to PLOS ONE should be reviewed and adhered to. the referencing styles are mismatched (with superscripts and numbered styles used). The numbers are placed after the punctuation marks (which should be for superscripts). Either way stick to author guidelines and be consistent.

Response: Thank you for highlighting this issue. We updated the references to adhere to the author guidelines provided by PLOS ONE.

2.

Comment 2: In the Eligibility criteria, provide further description for what will be considered screening behaviour to warrant exclusion.

Response: In our revised Methods (Stage 2: Study Eligibility, p. 7, lines 143–145), we now specify that screening behavior refers exclusively to the completion or uptake of any recommended CRC screening test (e.g., fecal occult blood test, sigmoidoscopy, colonoscopy) as confirmed either by self-report or medical records.

We further clarify that studies measuring only intermediate outcomes such as knowledge, attitudes, or intentions—with no data on actual screening completion—will be excluded (p. 8, lines 161-162).

3.

Comment 3: Also provide rationale for exclusion of review studies from inclusion.

Response: In our revised Methods (Stage 2: Study Eligibility, p. 8, lines 166-168), we have updated the language to say,

Title: Addressing the immigrant screening gap: A protocol for a systematic review on

interventions to enhance colorectal cancer screening among immigrants in the United States

Manuscript ID: PONE-D-24-23935

Date: 1/29/2025

“We exclude systematic reviews (and other review articles) from our final dataset because they do not provide original data on screening outcomes for our quantitative synthesis. However, in line with the best practices outlined by Cochrane, we review their reference lists to ensure we identify any potentially relevant original studies that we might have otherwise missed. This step helps us maintain a comprehensive search strategy without duplicating the reviews’ secondary data.”

4.

Comment 4: Would protocols be excluded too as you cite on line 214 "conducting searches forward from relevant, peer-reviewed research protocols to find articles discussing study results." Why is that?

Response: Yes, in our revised Methods (Stage 2: Study Eligibility, p. 8, lines 173-174), we have updated the language to say,

“We will also exclude articles that are conference abstracts or study protocols if they do not provide enough data to extract, are missing full text records, or don’t have results posted on ClinicalTrials.gov.”

Protocols, by definition, do not report results or outcome data on screening behavior. Because our systematic review focuses on interventions that have measured CRC screening uptake, we exclude protocols from the final inclusion. Thank you for pointing out that they should have been listed as an exclusion!

The forward citation search from protocols (line 214) is specifically intended to locate any subsequent publications that present the final study results. If such a publication exists and meets our eligibility criteria (e.g., reports screening uptake data), we will include that publication in our review. This procedure prevents us from missing relevant studies that were initially published only as protocols and later released with outcome data.

5.

Comment 5: What will be the grading system for consideration of moderate to good inter-rater agreement for your Kappa scoring?

Response: We will use the grading used in the cited article (Reference #38). We will consider 0.60-0.79 to be moderate level of strong. Strong will represent 0.80-0.90 and almost perfect will represent above 0.90. We have added the following text to p.11, lines 203-204, “We will consider 0.60 or higher as an acceptable level of inter-rater reliability.”

6.

Comment 6: From Line 211 to 215, I don't understand the rational for the 3 steps itemized. I find it redundant if you already did that in your screening. Except it is any different from what is already described I advise further explaining your rationale or removing it. For example, why will you conduct searches again in step (c) to find articles discussing results? and what is the point of reviewing bibliographies again when

Title: Addressing the immigrant screening gap: A protocol for a systematic review on

interventions to enhance colorectal cancer screening among immigrants in the United States

Manuscript ID: PONE-D-24-23935

Date: 1/29/2025

you already mentioned it in your eligibility criteria on line 164.

Response: Thank you for this comment. We agree that this piece is redundant and confusing, and we have removed this section.

7.

Comment 7: What would you use to appraise cross-sectional studies. ROBINS-I?

Response: Yes, we will use ROBINS-I! Clarified on p.13, lines 246-248 as below.

“The Risk of Bias in Nonrandomized Studies of Interventions (ROBINS-I) tool will be used for all other observational study designs (e.g., pre-post study designs, cohort studies, cross-sectional) [1].”

8.

Comment 8: Line 263: "If consensus cannot be reached after discussion with both reviewers, the supervising author will be consulted (AR)" is redundant. It has been described before in Line 204.

Response: Agreed. This paragraph was shortened to remove the redundancies. Thank you!

9.

Comment 9: Line 264:"We will report on themes that contribute to a higher risk of bias rating in the full review (i.e., unmasked outcome assessment, differential awareness of trial participation)." what does this mean? what is the aim vis a vis your core objectives.

Response: Thank you for seeking clarification on this point. By “themes that contribute to a higher risk of bias rating,” we are referring to specific methodological issues (e.g., unmasked outcome assessment, lack of blinding, or differential trial awareness among participants) identified through the ROBINS tools. It is standard when using these tools to report on common sources of bias that elevate a study’s risk rating, as doing so helps readers understand why a particular study may be classified at a higher risk of bias and how that classification might influence our overall conclusions regarding the effectiveness of the interventions under review.

10.

Comment 10: Line 228: "Missing data in publications that are requested on the extraction form will be elicited by contacting the listed contact author once by email at the email address listed on the publication." This is a bit vague and I am unsure if contacting corresponding authors would significantly benefit the design of your study, which relies on secondary data. If the purpose is to clarify already published data, this should generally be acceptable. However, if the intent is to collect additional data, such as missing information, this would likely require an ethics application for secondary use of data. I recommend confirming the ethical requirements with your institution to ensure compliance.

Title: Addressing the immigrant screening gap: A protocol for a systematic review on

interventions to enhance colorectal cancer screening among immigrants in the United States

Manuscript ID: PONE-D-24-23935

Date: 1/29/2025

Response: Thank you for your important feedback. We confirm that our sole purpose in contacting authors is to seek clarification of published data (e.g., sample sizes or screening rate percentages that appear inconsistent or are missing). We do not intend to collect any new or unpublished data. Rather, we will limit our inquiries to requesting clarifications on data that should already be reported in the manuscript. Given this scope, we do not anticipate ethical concerns regarding secondary data usage. We have updated the language in the Methods (Data Collection, p.12, lines 222–224): “If clarifications of published data are needed (e.g., sample size, reported screening rates), we will attempt to contact the corresponding author once by email at the address provided in the publication [28].”

---

## [Decision Letter · Decision Letter 1]

16 Mar 2025

Addressing the immigrant screening gap: A protocol for a systematic review on interventions to enhance colorectal cancer screening among immigrants in the United States

PONE-D-24-23935R1

Dear Dr. McCready,

We’re pleased to inform you that your manuscript has been judged scientifically suitable for publication and will be formally accepted for publication once it meets all outstanding technical requirements.

Kind regards,

Udoka Okpalauwaekwe, MD, MPH, PhD

Academic Editor

PLOS ONE

Additional Editor Comments (optional):

Reviewers' comments:

Reviewer's Responses to Questions

**Comments to the Author**

1. Does the manuscript provide a valid rationale for the proposed study, with clearly identified and justified research questions?

Reviewer #1: Yes

Reviewer #2: Yes

2. Is the protocol technically sound and planned in a manner that will lead to a meaningful outcome and allow testing the stated hypotheses?

Reviewer #1: Yes

Reviewer #2: Yes

3. Is the methodology feasible and described in sufficient detail to allow the work to be replicable?

Reviewer #1: Yes

Reviewer #2: Yes

4. Have the authors described where all data underlying the findings will be made available when the study is complete?

Reviewer #1: Yes

Reviewer #2: Yes

5. Is the manuscript presented in an intelligible fashion and written in standard English?

Reviewer #1: Yes

Reviewer #2: Yes

6. Review Comments to the Author

You may also provide optional suggestions and comments to authors that they might find helpful in planning their study.

Reviewer #1: Please considering citing one good papers that has an alluvial chart since you are proposing to use it for visual allustration.

Reviewer #2: Thanks for the opportunity to review this work again. I believe all comments seem to be sufficiently addressed. I have no further comments and look forward to reading your work in the nearest future.

7. PLOS authors have the option to publish the peer review history of their article (what does this mean? ). If published, this will include your full peer review and any attached files.

**Do you want your identity to be public for this peer review?** For information about this choice, including consent withdrawal, please see our Privacy Policy .

Reviewer #1: No

Reviewer #2: **Yes: ** Udoka Okpalauwaekwe MD, MPH, PhD

---

## [Editor Report · Acceptance letter]

PONE-D-24-23935R1

PLOS ONE

Dear Dr. McCready,

I'm pleased to inform you that your manuscript has been deemed suitable for publication in PLOS ONE. Congratulations! Your manuscript is now being handed over to our production team.

Kind regards,

on behalf of

Dr. Udoka Okpalauwaekwe

Academic Editor

PLOS ONE